# Design Methodology and Technology of Textile Footwear

**DOI:** 10.3390/ma15165720

**Published:** 2022-08-19

**Authors:** Patrycja Kaziur, Zbigniew Mikołajczyk, Magdalena Kłonowska, Bogusław Woźniak

**Affiliations:** 1Department of Knitting Technology and Textile Machines, Faculty of Materials Technologies and Textile Design, Lodz University of Technology, 90-924 Łódź, Poland; 2Łukasiewicz Network—Institute of Sustainable Technologies, 26-600 Radom, Poland

**Keywords:** footwear, comfort, textile packages, footwear upper, microclimate

## Abstract

Material selection is an important stage in the design of footwear, which determines not only its appearance, but also the comfort of use. The paper presents the methodology for the design of sports textile footwear manufactured by innovative hybrid technology. The upper part of the footwear, designed in CAD and a graphics program, consisted of two elements. Both of them were characterized by a three-layer knitted and embroidered structure. The air permeability and thermal insulation parameters were determined for each layer and for the whole package. The conducted tests demonstrated that the openwork spacer knitted fabric, as a component of the package, increased the air circulation inside the footwear as well as improved the heat transfer outside.

## 1. Introduction

The modern textile industry places great emphasis on innovation and sustainable development [1]. Manufactured textiles are an integral part of every area of human life. Currently, they are gaining more and more importance in the footwear industry. One of the technologies used in the production of footwear, especially sports, leisure, or specialist footwear, is the technology in which the upper part of the product is manufactured in the form of a fully-shaped textile element [2,3,4]. This solution allows for shaping the material and design of the footwear upper due to the use of fabrics with a heterogeneous (anisotropic) structure for the individual parts of the foot.

One of the well-known companies that design and improve the outer materials of sports footwear in terms of strength and functionality is Nike [3]. The brand has introduced the following collections: FlyKnit, FlyEasy, FlyWire, Metcon, LeBron, in which the uppers are manufactured in the form of fully-shaped knits. The technology developed by the company allows for reducing material waste by up to 80% compared to conventional production methods [5]. Additionally, polyester obtained from recycled plastic bottles is used in the manufacturing process. Similarly, Adidas developed its Primeknit technology, in which the footwear uppers are manufactured in the form of fully-shaped elements [3]. Such uppers are characterized by high durability, accompanied by optimum lightness, breathability, and flexibility.

Both technologies were developed using flat knitting machines produced by Stoll. The textile elements of footwear can also be produced on warp knitting machines and circular machines with large and small diameters. Karl Mayer is the leading manufacturer of warp knitting machines for the production of uppers [6]. Large-diameter cylindrical knitting machines are produced by Mayer & Cie, whereas small-diameter ones come mainly from the Santoni and Colosio companies [7,8,9].

The textile footwear design methodology demonstrates an increasing tendency to combine different technologies and materials and personalize the final product. For example, Alexander Taylor, a London-based designer, proposed a production alternative to Adidas—the Futurecraft Tailored Fiber model, based on tailored fiber placement (TFP) technology [10]. The designer used embroidery technology to program the pattern and the course of stitches to meet the requirements of individual users. In this model, the upper is created by combining different materials to form a fully-shaped element with different properties and strengths for the selected areas of the foot. Such a design ensures adequate support and comfort of use.

The foot is one of the parts of the human body that does not maintain constant temperature [11,12]. The lowest temperature values are observed for the toes, while the highest ones appear in the metatarsal and tarsal areas. The temperature in the footwear microclimate, which is most comfortable for the user, ranges from 28 to 34 °C [13]. The relative humidity should, however, be maintained at the level of 60 to 65% [14]. Even slight deviations from these values may cause discomfort and lead to hypothermia or overheating of the body. These processes are closely related to heat exchange with the environment as well as the production of sweat and its transfer away from the surface of the skin; therefore, according to many sources, thermo-physiological comfort is one of the most important aspects determining the functionality of footwear [15,16,17].

The optimum temperature flow inside footwear can be achieved by selecting a set of textile materials that are appropriate in terms of thermal insulation, paying the greatest attention to the volumetric weight, thickness, porosity, and thermal conductivity of fibers [17,18]. An example of thermal comfort modeling in footwear is the use of membranes, where the outer layer and the lining are matched to the supporting material [19].

Many scientific papers have confirmed the legitimacy of using multilayer compositions to improve the functionality of footwear [12,20]. The conducted analysis confirmed the importance of the relationship between the different systems of outer materials and linings for modeling the comfort properties of footwear [16,21]. The optimal choice of footwear materials can minimize the discomfort conditions inside the footwear. The analysis of mechanical and hygienic properties of the selected knitted systems based on natural fibers combined with synthetic fabrics confirms that their use as footwear materials can significantly affect the quality of use of the final product [22].

The aim of the work was to develop hybrid technology for the outer part of textile footwear (uppers), along with the modeling of functional comfort in terms of thermal and physiology. The partial goal of hybrid technology combines 2D and 3D knitting technology with the embroidery technique. The created multilayer material package of the uppers as well as its components were subjected to functional tests of thermo-physiological comfort. The purposefulness of this research was dictated by the presentation of the production of a new technology of the textile elements of footwear as well as the characterization of their comfort parameters.

In the design of the textile footwear elements, it was intended to create an upper in the form of a multilayer knitted and embroidered structure, where knitted fabrics were used for the inner layers to ensure physiological and sensory comfort, while the outer layer was made of embroidered polyester threads resistant to mechanical factors and characterized by certain design aesthetics. A standard model of the foot, the so-called ‘last’, and the patterns made available by the RenBut footwear manufacturing company, were used in the CAD design of the shoe upper form. The project was implemented in the form of a finished pair of sports footwear, manufactured in industrial conditions, based on the developed 3D form, material composition, and technology. Air permeability and thermal resistance with the associated parameters were determined for the materials used in the project and their packages.

## 2. Materials

The objective of the project was to produce the footwear using hybrid technology, combining knitted materials with embroidery. Subsequently, raw materials were carefully selected to increase the footwear comfort. It was decided to use a three-layer structure with improved biophysical properties, combining synthetic and natural fibers. A polyester spacer knitted fabric (I) was selected for the inner layer in contact with the foot (the so-called lining), a cotton weft-knitted fabric (II) was proposed for the adjacent layer, while embroidery with recycled polyester threads (III) was chosen for the outer layer of the upper. The schematic structure of the footwear is presented in Figure 1.

In the construction of footwear, the structure of embroidery is also important. It consists of two types of threads intertwined with each other through the base layer, formed by cotton weft knitted fabric (II). The top thread is a polyester embroidery thread, and the bobbin thread is made of cotton (Figure 2).

In the design of footwear, it was decided to use a multi-layer system of materials by combining synthetic and natural materials, following the canon of construction of a two-layer knit with increased biophysical values [20]. According to this canon, a polyester spacer knitted fabric with hydrophobic properties (i.e., low moisture absorption rate), constitutes the footwear lining, that is, the conductive diffusion layer adjacent directly to the body. The moisture appearing inside the footwear is transferred to the next sorption layer with hydrophilic properties, made of cotton knitted fabric, which is excluded from direct contact with the user’s skin. The adopted material design ensures an appropriate microclimate inside the footwear. The structure of the spacer knitted fabric, which consists of two different outer layers (smooth and openwork) connected by a monofilament layer also contributes to the increased physiological and hygienic comfort (Figure 3) [23]. The smooth layer is directed toward the cotton fabric, while the openwork is in contact with the foot. The 3D knitted fabric provides very good air permeability and minimal absorption of moisture and water. Its porous structure in direct contact with the skin allows for air access to the foot, improving the ventilation of the footwear. In order to give the footwear additional cushioning properties and air pumping function, knitted fabric with a structure different from that of the lining can also be used as an insert. The 3D knitted layer also minimizes the growth of bacteria and mites.

The spacer knitted fabric (I) used for the lining was designed in the ProCad Warpknit program, in which its structure was visualized (Figure 4) The outer layer, marked in blue, with an openwork surface, is created from two component stitches made by GB1 and GB2 needle bars with incomplete threading. The inner layer made of monofilament yarn (green) is created by GB3 and GB4 needle bars. The GB5 and GB6 needle bars are responsible for creating the last outer layer with a smooth surface made of two-component stitches, marked in red.

For each of the knitted fabrics, photographs of the right and left sides were taken using OPTA-TECH’s X2000 stereoscopic microscope (Warsaw, Poland). For the 3D fabric, cross-sectional photos were also taken (Figure 5).

A cotton interlock knitted fabric with pique-type stitches was used for the layer adjacent to the spacer knitted fabric due to its strong hygroscopic properties and high absorbency, accompanied by strength (Figure 6).

The outer layer of the upper consists of embroidery threads made of continuous polyester fibers. In the cross-section, they have a trilobal shape, increasing the area reflecting the sun rays, which is a desirable phenomenon, especially during summer when temperatures are high. The polyester threads were selected due to their functionality, color durability, abrasion resistance as well as resistance to washing and mechanical damage. The embroidery was mainly made with a filling stitch, the so-called tatami, which consists of a large number of punctures, as a result of which the outer structure of the footwear is characterized by dense filling (cover) of the upper surface with threads, high durability, and deformation resistance.

Table 1 describes the type of fabric, the raw material, the yarn used, and the weave (in the case of the third outer layer—stitch) for each of the layers of the upper.

The basic structural parameters determined for the knitted fabrics used in the footwear are summarized in Table 2.

## 3. Methods

### 3.1. Longitudinal Strength of the Materials Used for Footwear

The longitudinal strength test was carried out for the reference materials used in shoes. Reference samples were cut from the knitted upper from the toe and midfoot areas (Figure 7) and tanned shoe leather (Figure 8). The knitted fabric tested had a mass per square meter equal to 1003.83 g/m^2^ and a thickness equal to 2.46 mm. The number of courses per decimeter for the toe area was 95 and the number of wales per decimeter was 65, while the number of courses per decimeter for the midfoot area was 100 and the number of wales per decimeter was 70. The tested leather had the following physical parameters: mass per square meter 1080.50 g/m^2^ and thickness of 1.45 mm.

The longitudinal strength test was carried out in accordance with the EN ISO 13934-1:2013 standard [24]. The results of the study are summarized in Table 3 and presented in the form of a graph (Figure 9).

From the presented data, it can be seen that the knitted elements of the footwear upper were two to four times more durable than the traditional leather material of sports shoes of a well-known European manufacturer.

### 3.2. Methods of Designing the Upper of Footwear

The upper part of the footwear was designed based on the templates of the sports shoes shown in Figure 10 and a size 33 last (Figure 11) provided by the Polish shoe company RenBut [25]. The number of textile elements was reduced from ten to two, dimensioned, and designed in the ZW3D CAD program. ZW3D CAD is an integrated CAD/CAM program that allows 3D designs of any shape to be obtained. The tools available in the software also allow for the creation of 2D drawings and compilations, together with the list of materials used. The TRShoeMaker program is used in the footwear sector to design and model different types of footwear. In this case, textile elements were designed in the form of two-dimensional objects (Figure 12).

The design projects of the shoe, taking into account the anatomical structure of the entire foot and its areas, were obtained in Corel Draw (Figure 13). A personalized logo, reflecting the initials of the Lodz University of Technology, was also added.

### 3.3. Technologies Used to Obtain the Third Layer of the Upper of the Footwear

The embroidery with polyester threads was partly made at the Department of Knitting Technology and Textile Machines on an FT-1201+1PD single-head computerized embroidery machine manufactured by the Fortever company (Figure 14). The Wiwa embroidery workshop equipped with a TMFD-G1215 12-headed machine “Tajima” (Figure 15) was also engaged in cooperation. The Fortever computerized embroidery machine has a working area of 560 × 1200 mm and a maximum speed of up to 1200 stitches per minute. The Tajima embroidery machine has a working area of 400 mm × 660 mm and a speed of up to 800 stitches per minute. Both machines use sharp type (RG) needles, size 80/12. The course of the embroidery threads was designed in the machine-dedicated emCAD and DG 16 by Pulse programs.

In the embroidery design, two types of filling stitches were used: tatami and satin. The tatami stitch, which owes its name to the structure of Japanese mats, is mainly used in computer embroidery to fill larger areas. The satin stitch, on the other hand, is used for smaller elements, inscriptions, and borders. The length of the thread between the stitches ranged from 0.43 to 0.47 cm, while the density of the thread arrangement in the embroidery was 40 threads per 1 cm of embroidery width. For both elements, five different colors were selected. The total number of stitches for the entire embroidery was 70,660. The thread course for each shoe element obtained in the DG by the Pulse program is presented in Figure 16. The design also included some non-embroidered areas to be removed, which together with the leatherwear eyelets serve as shoelace holes (Figure 17).

### 3.4. Footwear Manufacturing Technology in Industrial Conditions

After the process was completed, the textile uppers were transferred to the RenBut company to manufacture the finished footwear (Figure 18).

Table 3 shows the carousel rotation. A special leather insole with antibacterial properties, appropriate for the shoe size, was inserted into the finished footwear (Figure 19).

In the newly designed footwear, a spacer knitted insert will be used, instead of the traditional leather insole (Figure 20), to increase the moisture flow and intensify the air ventilation.

### 3.5. Test Methods

For the cotton weft-knitted fabric marked with II and the following packages, weft-knitted fabric with embroidery (II + III) and spacer knitted fabric and weft-knitted fabric with embroidery (I + II + III), and air permeability R was determined according to the EN ISO 9237:1995 standard [26]. In contrast, the measurement of air permeability for the spacer lining fabric (I) was carried out according to the proprietary method on a device patented by the Lodz University of Technology [27]. For each of the knitted fabrics and their packages with embroidery, some selected thermal insulation parameters were also determined using the Alambeta device. These were the basic parameters of the footwear material components, which determine the physiological comfort of the user.

#### 3.5.1. Air Permeability Test

The measurement of air permeability according to EN ISO 9237:1995 was carried out on an instrument equipped with a sample jamming holder and a system enabling differential pressure to be obtained on both sides of the sample, with the possibility of measuring its value on the pressure gauge. For each of the tested variants, 10 measurements were performed at various locations of the sample. The pressure drop used was 200 Pa (technical products) and the tested surface was 9.616 cm^2^. The study was conducted under normal climate conditions. The test was carried out at a temperature of 20 °C and relative humidity of 65%. Air permeability (*R*) was calculated from the formula:$$R=\frac{{\bar{q}}_{v}}{A}\times 167\;$$
where:

${\bar{q}}_{v}$—arithmetic mean of the amount of air flowing, dm^3^/min,

*A*—tested surface of the sample, cm^2^,

167—conversion factor.

Due to the high porosity of the spacer knitted fabric and the impossibility of reading differential pressure on both sides (I), a permeability test was performed according to a different method. For this purpose, a device for measuring the air permeability of high-porosity 3D materials described in Polish patent PL 237,156 was used [27]. The instrument (Figure 21) consists of a measuring head (1), a pipeline (3), an equalizer tank (4), a suction pipe (5), a device forcing air flow through the sample (6), and a vertically located section of the piping (7).

Samples of the 3D knitted fabric (I) with a diameter of 5 cm were successively fixed in the holder of the open measuring head, after which the suction was actuated, and the air pressure drop was measured after passing through the material sample in a perpendicular direction. Ten measurements were made at a pressure drop of 200 Pa (technical products). Air permeability was determined according to the same dependence as for the other materials.

#### 3.5.2. Thermal Insulation Test

A thermal insulation test of the footwear material layers was carried out according to the methodology of the Alambeta instrument (Figure 22), which is a double-plate device using the constant heat transfer phenomenon. This method involves measuring the amount of heat flowing through the sample placed between the two plates. The upper plate simulating the temperature of human skin is heated to 32 °C, while the lower plate maintains the normal climate temperature (about 20 °C). The instrument allows for the determination of the static and dynamic thermal insulation parameters of textile products. The study was conducted under normal climate conditions (temperature: 20 °C, relative humidity: 65%).

The parameters determined with the use of the Alambeta instrument are presented with their definitions and units in Table 4.

## 4. Results

### 4.1. Air Permeability

The air permeability results obtained according to EN ISO 9237:1995 and the results obtained for the spacer knitted fabric according to the patented method are included in Table 5.

The highest air permeability value was obtained for the spacer knitted fabric used as a shoe lining, while the lowest was for the cotton weft-knitted fabric with embroidery (package II + III). Based on the presented results, it can be concluded that the embroidery layer significantly reduces the air permeability of the tested material. Some improvement of the footwear comfort was achieved by using an openwork 3D knitted fabric with increased air circulation. Due to the structure of the spacer knitted fabric, air transport takes place not only in the perpendicular direction to the fabric surface, but also along the upper due to the so-called ‘chimney effect’. In the case of the designed footwear, this is a positive phenomenon. The spacer knitted compartment between the user’s foot and the outer layer of the upper improves the footwear ventilation, thus increasing the moisture transport from the layer next to the skin to the environment. This phenomenon is intensified by the so-called ‘pumping’ effect due to movements of the foot inside the shoe.

### 4.2. Thermal Insulation

The obtained results of thermal insulation are presented in Table 6 and Table 7.

The thermal conductivity (λ) of a material is a measure of its ability to conduct heat. It is expressed as the amount of heat transferred in a unit of time by a unit of surface, at a temperature decrease of one K per unit of the tested material thickness. The lowest conductivity value was obtained for the polyester spacer knitted fabric (I) and the highest for the package containing all of the tested components (I + II + III).

Thermal diffusion (a) is the material property that characterizes heat conduction under undetermined conditions. It allows determining the reaction rate of the tested material to temperature changes. The openwork spacer knitted fabric (I) was characterized by the highest value of thermal diffusion, while the lowest one was obtained for the weft-knitted fabric with embroidery (II + III). Based on the analysis of the results obtained, it can be concluded that the spacer knitted variant is characterized by the highest ability to transfer heat. Applying such a fabric in the package, together with other materials of the shoe upper, will improve heat transport from the inside of the footwear to the environment.

Thermal absorption (b) is characterized by the amount of heat carried by a unit of surface in a unit of time, at a temperature decrease of one K per unit of volume. The highest thermal absorption value was determined for the weft knitted fabric with polyester thread embroidery, while the lowest was obtained for package I + II + III.

Thermal resistance is the ratio of the material layer thickness to its heat conductivity coefficient. The greater the thickness of the tested material layer, the higher the thermal resistance it demonstrates. The lowest value of thermal resistance was demonstrated by the weft knitted fabric, which was at the same time the thinnest of the tested materials. On the other hand, the highest value was recorded for the package containing all layers. Taking into account the results obtained, it can be concluded that the package I + II + III will be characterized by the best heat insulation properties. Allowing for the diffusion parameters, the results demonstrate that introducing spacer knitted fabric into the package improves its capability of heat transfer outside. In the case of summer shoes, it is possible to apply the uppers in the form of an openwork structure, which will improve the ventilation of the footwear (human foot) in terms of both increased heat and moisture transmission.

When analyzing the results obtained for the tested materials and their packages, it should be taken into account that the Alambeta device is designed to measure the thermal resistance by conduction, in tests carried out under ambient conditions.

To relate the results of the obtained values of thermo-physiological parameters of the new hybrid technology, the selected parameters of the traditional reference material were tested for comparison. The tested material was tanned cowhide, for which the average air permeability was 0.4 mm/s, and the average thermal resistance was 26.9 × 10^−3^ W^−1^·K·m^2^. The material package used in the designed upper showed more than a hundred times better permeability than the reference leather material. For the hybrid technology, the thermal resistance value was also three times higher, thanks to which the designed footwear will be characterized by better thermal insulation.

## 5. Summary and Conclusions

### 5.1. Summary

One of the technologies used in manufacturing footwear, especially sports, leisure, or specialized, is the technology in which the upper part of the product is produced in the form of a fully-shaped textile element. This method makes it possible to shape the material and design of the uppers, together with programming their mechanical, sensory, and physiological properties. The world-famous Nike and Adidas companies tend to use fully-shaped weft-knitted fabrics for the outer elements of footwear uppers, which ensures high mechanical strength accompanied by lightweight, breathability, and flexibility. Today, knitting technologies are the only potential techniques for manufacturing textile footwear in the world. In the implementation of the project, the effects of which are presented in this publication, it was assumed to produce the textile elements of the footwear uppers in the form of multi-layered structures.

One of the objectives of the project was to produce the footwear in hybrid technology, combining knitted materials with embroidery. Raw materials increasing the comfort of footwear use were selected. It was decided to use a three-layer structure with improved biophysical properties, combining synthetic and natural fibers. A polyester spacer knitted fabric was selected for the inner layer coming into contact with the foot (the so-called lining), a cotton weft-knitted fabric for the adjacent layer, whereas embroidery with recycled polyester threads was chosen for the outer layer of the upper. In such layered structures, the polyester spacer knitted fabric with good hydrophobic properties, that is, low moisture absorption rate, constitutes the lining of the footwear (i.e., a conductive diffusion layer adjacent directly to the body). The moisture appearing inside the footwear is transferred to the next sorption layer with hydrophilic properties, which is made up of the cotton knitted fabric. The adopted material structure ensures an appropriate microclimate inside the footwear. The shoe upper was designed in the ZW3D CAD program using the additional TRShoeMaker function in the form of two-dimensional objects. The embroidery was made on a twelve-head TMFD-G1215 Tajima machine with a 400 mm × 660 mm working area and a speed of up to 800 stitches per minute. The course of the embroidery threads was designed in the DG by the Pulse program dedicated to the machine. The length of the thread between the stitches ranged from 0.43 to 0.47 cm, whereas the density of the thread arrangement in the embroidery was 40 threads per 1 cm width of the embroidery. After that, the textile uppers were transferred to the RenBut company to produce the finished footwear.

For the material components of the upper: cotton weft-knitted fabric, packages of weft-knitted fabric with embroidery, and packages of spacer knitted and weft-knitted fabrics with embroidery, the air permeability R was determined according to the EN ISO 9237:1995 standard. For the spacer knitted lining, the air permeability measurements were carried out according to the proprietary method on a device patented by the Lodz University of Technology. For each of the knitted fabrics and their package with embroidery, the selected thermal insulation parameters were also determined with the use of the Alambeta instrument. These are the basic parameters determining the physiological comfort of the footwear user. The highest air permeability (i.e., 3322 mm/s) was obtained for the spacer knitted footwear lining, while the lowest of 133 mm/s was observed for the cotton weft-knitted fabric with embroidery. Using openwork 3D knitted fabric with increased air circulation improved the comfort of footwear use. Due to the structure of the spacer knitted fabric, air transport takes place not only in the direction perpendicular to the fabric surface, but also along the upper, due to the so-called ’chimney effect’. The spacer knitted compartment between the user’s foot and the outer layer of the upper improves the footwear ventilation, thus increasing the moisture transport from the layer next to the skin to the environment. This phenomenon is intensified by the so-called ’pumping‘ effect due to foot movements inside the shoe. The lowest thermal resistance value was demonstrated by the weft-knitted fabric, which was at the same time the thinnest of the tested materials, whereas the highest value was recorded for the package containing all of the tested layers. Allowing for the diffusion parameters, the results demonstrate that introducing a spacer knitted fabric into the package improves its capability of heat transfer outside. In the case of summer shoes, it is possible to apply the uppers in the form of an openwork structure that improves the ventilation of the footwear (human foot) in terms of both increased heat and moisture transmission.

The study of the functional properties of the textile footwear was limited to three important basic parameters, which are longitudinal strength, thermal insulation, and air permeability. Many other tests could be performed, for example, the alternating bending strength, tear strength, resistance to attachment of the sole, susceptibility to abrasion, pilling, resistance to wet alkaline or acid agents, resistance to washing, resistance to dirt, resistance to UV radiation, aging, and physiological evaluation by the artificial skin method and others. A comprehensive multi-parameter assessment of the selected variants of textile footwear will be presented in the following publications.

### 5.2. Conclusions

(1)The original structure of the upper of the footwear was developed in the form of a multi-layer knitted-embroidered structure, where a spacer knitted fabric was used for the layer adjacent to the foot, ensuring physiological-sensory comfort; the inner layer was made of cotton knit, and the outer layer was embroidered with polyester threads, which came from recycling. In the footwear construction project, it was decided to use a multi-layer system of materials combining synthetic and natural materials, following the canon of building a two-layer knit with increased biophysical values.(2)The analysis of the structural parameters showed that the materials used had a thickness ranging from 1.28 mm to 3.43 mm, and a mass per unit area ranging from 291.87 g/m^2^ to 321.84 g/m^2^. The parameters of the density were as follows: the number of courses per decimeter from 130 to 150, the number of wales per decimeter from 100 to 130.(3)Air permeability and selected thermal insulation parameters, which determine physiological comfort in the context of footwear use, were determined for the variants of cotton weft knit and packages: weft knit with embroidery and distance knit and weft knit with embroidery.
The highest value of air permeability equal to 3322 mm/s was obtained for the distance knitted fabric, which is the lining of the footwear, and the lowest for the cotton weft knitted fabric with embroidery equal to 132 mm/s. Based on the presented results, it can be seen that the embroidery layer significantly reduced the air permeability of the tested material. Due to the spatial structure of the spacer knitted fabric, air is transported not only perpendicular to the knitted fabric surface, but also along the upper, caused by the so-called ‘chimney effect’.The lowest value of thermal resistance equal to 28.5 × 10^−3^ W^−1^·K·m^2^ showed the weft knitted fabric, which was also the thinnest of the tested materials. The highest value of 92.4 × 10^−3^ W^−1^·K·m^2^ was recorded for the package of all layers. Taking into account the obtained results, it can be concluded that the three-layer package of materials will be characterized by the highest thermal insulation among the materials tested, proving that the share of the spacer fabric in the package will improve the ability to dissipate heat to the outside.The material package used in the designed upper showed over one hundred times better air permeability than the reference leather material. For the hybrid technology, an over three times higher value of thermal resistance was also obtained, thanks to which the designed footwear will be characterized by higher thermal insulation.(4)In the future, a new design methodology will be assessed against the use of different raw materials in a shoe material package to improve the physiological comfort. Therefore, technology can be used in the personalization of footwear.

## Figures and Tables

**Figure 1 materials-15-05720-f001:**
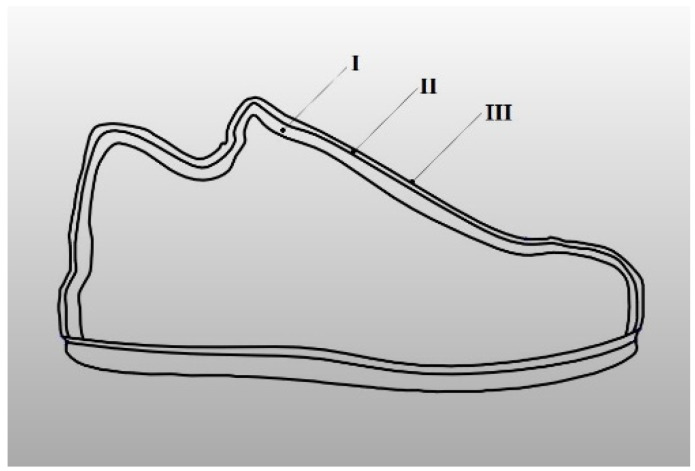
A schematic cross-section of footwear with marked layers (I—polyester spacer knitted fabric, II—cotton weft-knitted fabric, III—embroidery with recycled polyester threads).

**Figure 2 materials-15-05720-f002:**
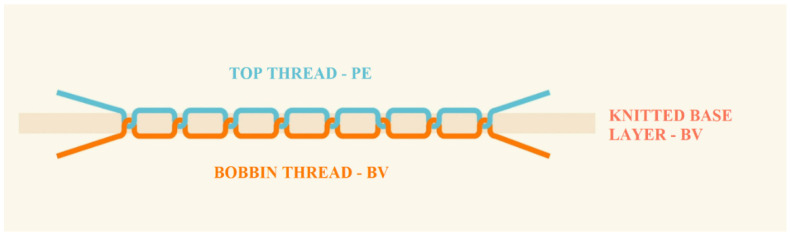
Schematic of the embroidery structure.

**Figure 3 materials-15-05720-f003:**
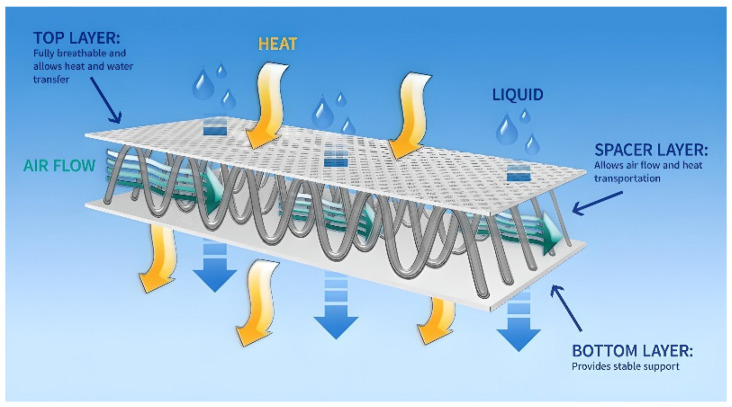
The structure of the spacer knitted fabric.

**Figure 4 materials-15-05720-f004:**
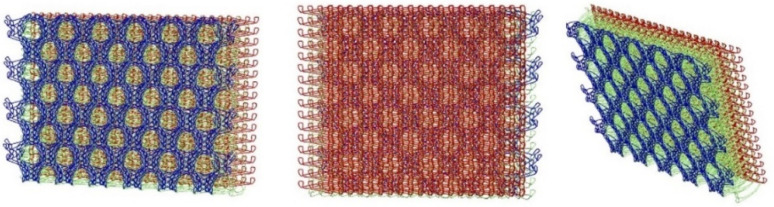
The visualization of spacer knitted fabric (I) designed in ProCad Warpkni.

**Figure 5 materials-15-05720-f005:**
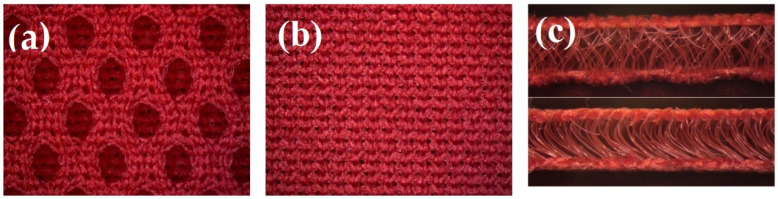
The right side of the spacer knitted fabric forming lining—I (**a**), left side of spacer knitted fabric forming lining—I (**b**), cross-sections of spacer knitted fabric forming lining—I (**c**).

**Figure 6 materials-15-05720-f006:**
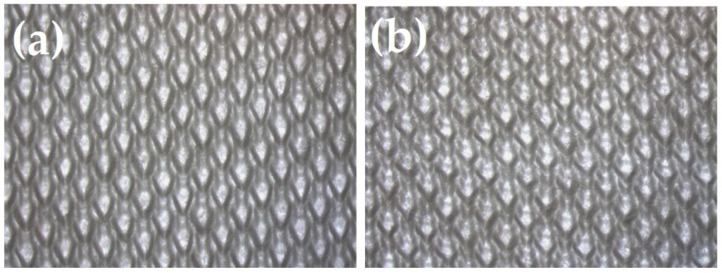
The right side of cotton weft-knitted fabric forming inner layer—II (**a**), left side of cotton weft-knitted fabric forming inner layer—II (**b**).

**Figure 7 materials-15-05720-f007:**
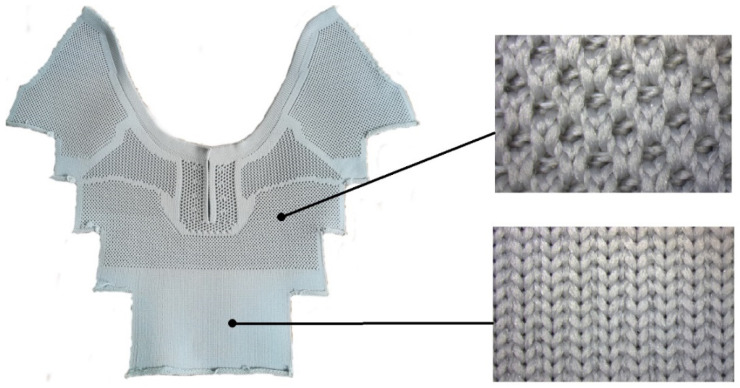
A fully-shaped shoe upper made using a Stoll flat knitting machine.

**Figure 8 materials-15-05720-f008:**
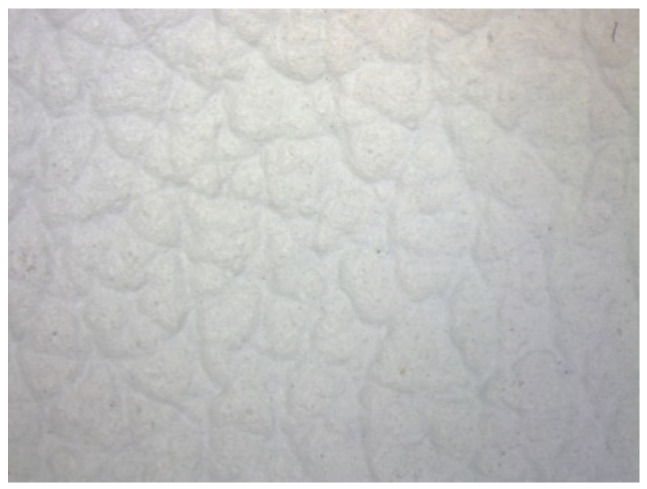
Tanned leather from the sports footwear.

**Figure 9 materials-15-05720-f009:**
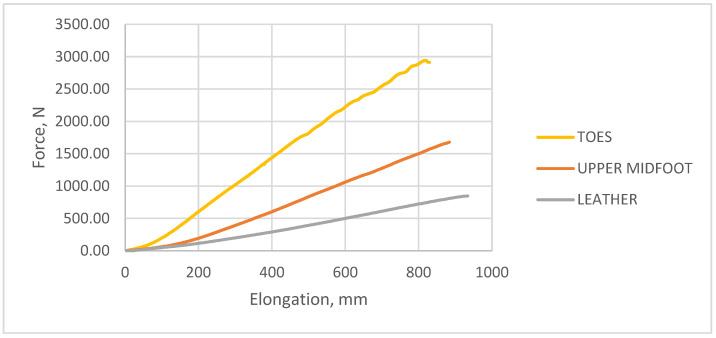
The tensile strength results for the selected footwear materials.

**Figure 10 materials-15-05720-f010:**
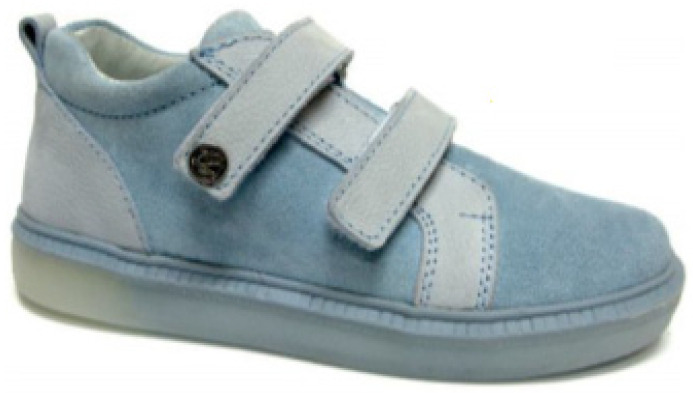
A model of the sports and leisure footwear.

**Figure 11 materials-15-05720-f011:**
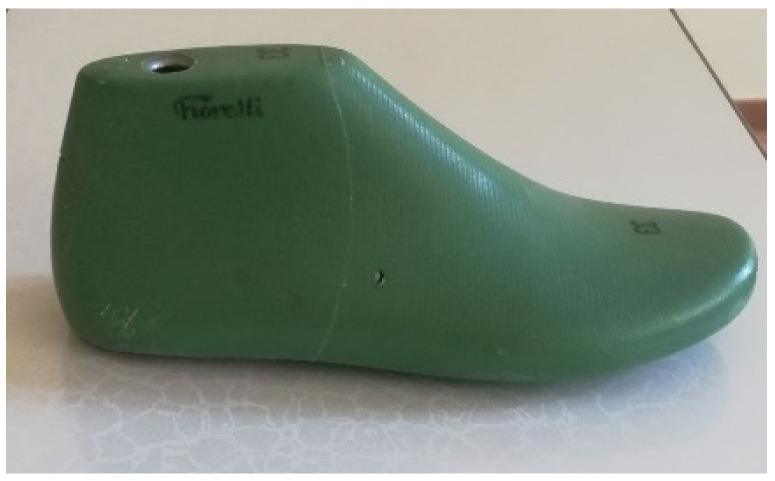
The last.

**Figure 12 materials-15-05720-f012:**
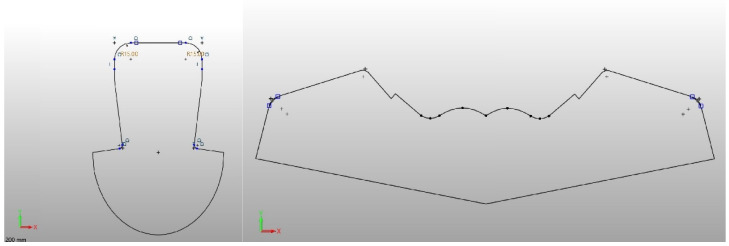
The template shapes of the footwear elements designed in the ZW3D CAD program.

**Figure 13 materials-15-05720-f013:**
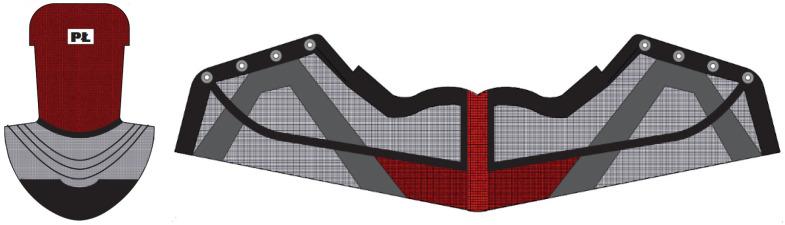
The design projects of the shoe elements made in Corel Draw.

**Figure 14 materials-15-05720-f014:**
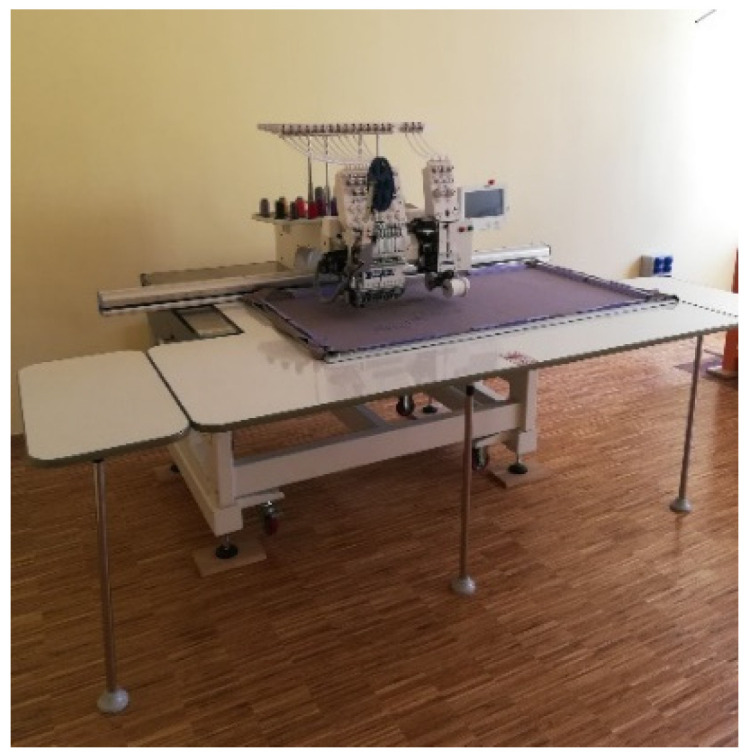
The FT-1201+1PD computer embroidery machine, Fortever.

**Figure 15 materials-15-05720-f015:**
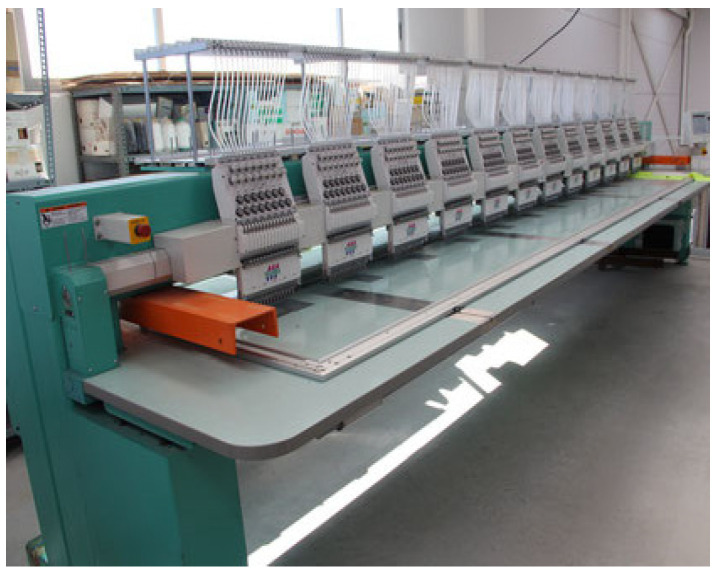
The TMFD-G1215 computer embroidery machine, Tajima.

**Figure 16 materials-15-05720-f016:**
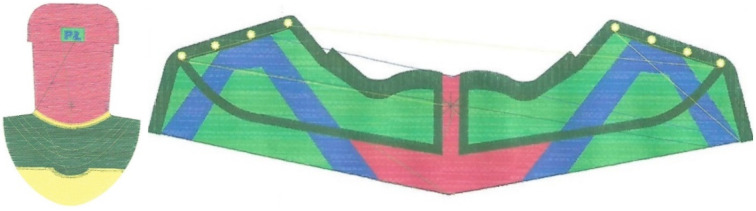
The embroidery design for the footwear elements obtained in the DG program by Pulse.

**Figure 17 materials-15-05720-f017:**
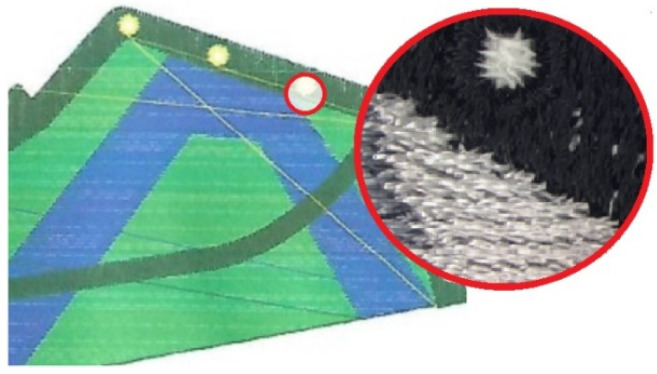
Part of the obtained embroidery.

**Figure 18 materials-15-05720-f018:**
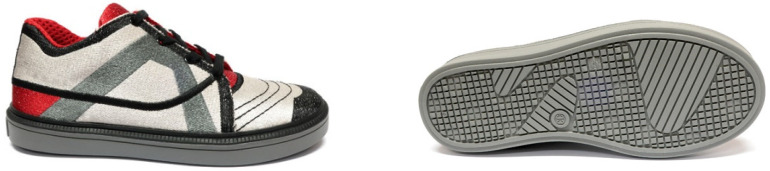
The sports footwear produced in industrial conditions.

**Figure 19 materials-15-05720-f019:**
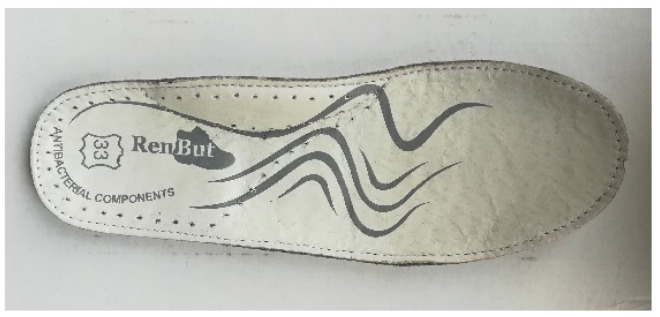
The antibacterial leather insert.

**Figure 20 materials-15-05720-f020:**
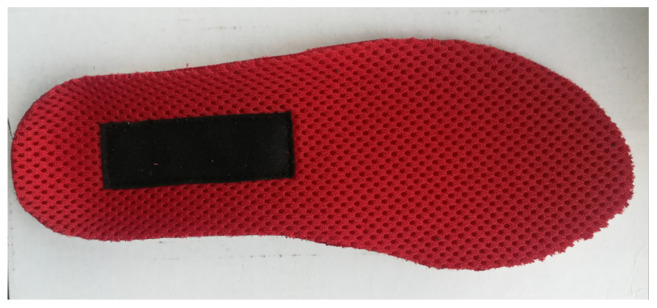
The footwear insert made of spacer knitted fabric.

**Figure 21 materials-15-05720-f021:**
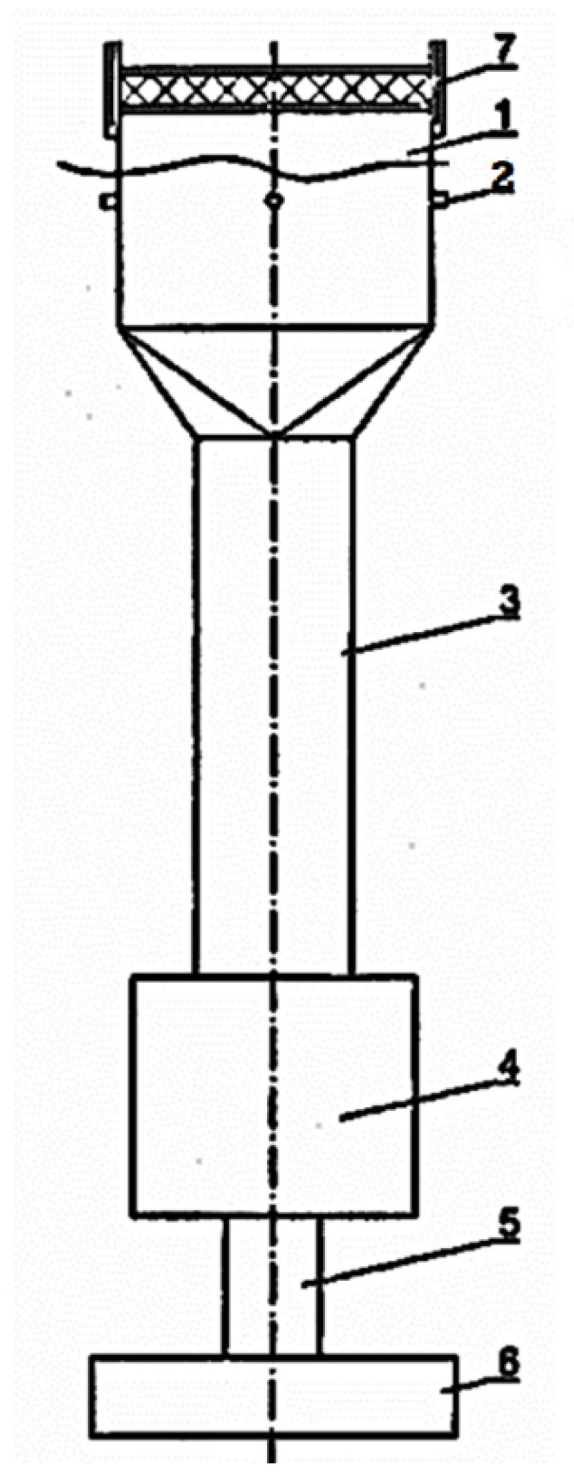
A schematic diagram of the device for measuring the air permeability of high-porosity 3D materials, (1—measuring head, 2—placement of the pressure drop device, 3—pipeline, 4—equalizer tank, 5—suction pipeline, 6—device forcing air flow through sample, 7—pipe section).

**Figure 22 materials-15-05720-f022:**
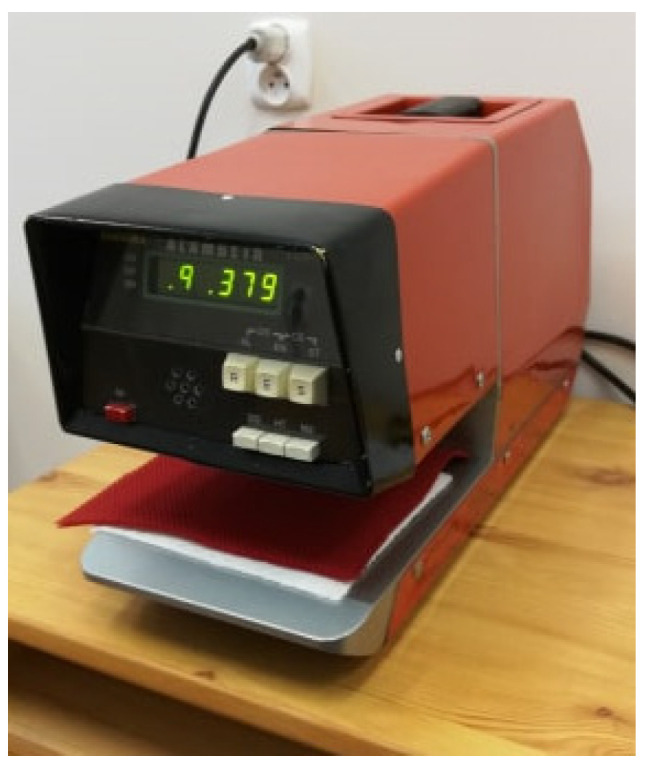
The double-plate Alambeta device.

**Table 1 materials-15-05720-t001:** The fabrics and raw materials used in the production of footwear.

Layer	Fabric Type	Raw Material	Yarn	Weave/Stitch
I	Spacer warp-knitted fabric	100% polyester	Outer layers: 110 dtex f24 Inner layer: Monofilament d = 0.06 mm	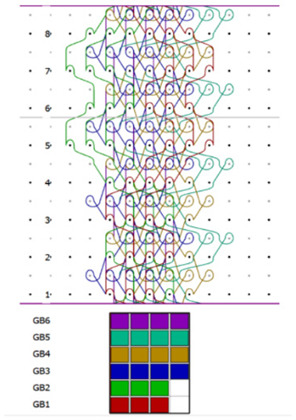 **GB1:** 3-4-4-4/3-2-2-2/3-4-4-4/3-2-2-2/1-0-0-0/1-2-2-2/1-0-0-0/1-2-2-2/ **GB2:** 1-0-0-0/1-2-2-3/1-0-0-0/1-2-2-2/3-4-4-4/3-2-2-2/3-4-4-4/3-2-2-2/ **GB3:** 1-2-2-3/2-1-1-0/1-2-2-3/2-1-1-0/1-2-2-3/2-1-1-0/1-2-2-3/2-1-1-0/ **GB4:** 2-1-1-0/1-2-2-3/2-1-1-0/1-2-2-3/2-1-1-0/1-2-2-3/2-1-1-0/1-2-2-3/ **GB5:** 3-3-3-4/1-1-1-0/3-3-3-4/1-1-1-0/3-3-3-4/1-1-1-0/3-3-3-4/1-1-1-0/ **GB6:** 0-0-0-1/1-1-1-0/0-0-0-1/1-1-1-0/0-0-0-1/1-1-1-0/0-0-0-1/1-1-1-0/
II	Weft-knitted fabric	100% cotton	20 tex yarn	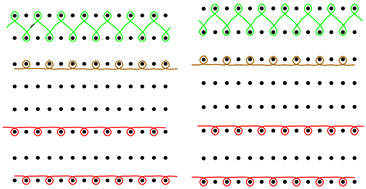
III	Embroidery	Top thread: 100% polyester Bobbin thread: 100% cotton	Top thread: 27 tex embroidery thread Bobbin thread: 40 tex embroidery thread	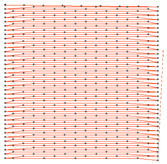

**Table 2 materials-15-05720-t002:** The characteristics of the knitted fabrics.

Fabric Symbol	Thickness [mm]	Standard Deviation	Mass per Square Meter [g/m^2^]	Standard Deviation	Density				Flat Loop Dimensions		Loop Shape Coefficient
					Course Density [Number of Courses/dm]	Standard Deviation	Wale Density [Number of Wales/dm]	Standard Deviation	Width [mm]	Height [mm]	
I	3.43	0.01	321.84	2.83	150	0.54	100	0.04	1.00	0.67	0.67
II	1.28	0.01	291.87	1.81	130	0.70	130.00	0.03	0.77	0.77	1.00

**Table 3 materials-15-05720-t003:** A summary of the tensile strength tests conducted according to the EN ISO 13934-1:2002 standard.

Material Tested	Maximum Breaking Force, [N]	Relative Elongation at Break, [%]
Knitted upper—toe area	2944.00	203.80
Knitted upper—midfoot area	1680.00	176.60
Leather	847.00	27.96

**Table 4 materials-15-05720-t004:** A summary of the parameters measured on the Alambeta instrument.

Parameter	Formula	Unit
Thermal conductivity, λ	λ = q(grad·t)^−1^	W·m^−1^·K^−1^
Thermal diffusion, a	a = λ·(ρ·c)^−1^	m^2^·s^−1^
Thermal absorption, b	$b=\sqrt{\lambda \cdot \rho \cdot c}$	W·m^−2^·s^1/2^·K^−1^
Thermal resistance, r	r = Q·Δt^−1^ = hλ^−1^	W^−1^·K·m^2^
Sample thickness, h	H	Mm
Max and stationary heat flow quotient, I	I = q_max_·q_s_^−1^	-
Stationary heat flow density at contact point, s	s = b·Δt·(π·t)^−0.5^	W·K·m^−2^

The symbols adopted in Table 4: t—temperature (K); q—heat flux density (W·m^−2^); c—specific heat (J·kg^−1^·K^−1^); Q—heat (J); ρ—density (kg·m^−3^); Δt—temperature difference on both sides of the sample.

**Table 5 materials-15-05720-t005:** A summary of the air permeability results according to the EN ISO 9237:1995 standard and the patented method for the spacer knitted fabrics.

Symbol of Knitted Fabric or Package	Mean Air Permeability $\bar{R}$, [mm/s]	Standard Deviation, σ	Air Permeability Variability Coefficient V, [%]	Confidence Interval at 95% Probability, [mm/s]
I	3321.980	3.79	0.11	0.13
II	1325.684	7.53	0.6	0.40
II + III	132.683	3.44	2.6	0.59
I + II + III	426.965	7.60	1.8	0.72

**Table 6 materials-15-05720-t006:** A summary of the parameters measured on the Alambeta device.

Symbol of Knitted Fabric or Package	Sample Thickness, h	Standard Deviation of Thickness σ_h_	Thermal Conductivity, Λ	Standard Deviation of Thermal Conductivity σ_Λ_	Thermal Diffusion, A	Standard Deviation of Thermal Diffusion σ_A_	Thermal Absorption, b	Standard Deviation of Thermal Absorption σ_b_
	[mm]	-	[10^−3^ W·m^−1^ ·K^−1^]	-	[10^−6^ m^2^·s^−1^]	-	[W·m^−2^·s^1/2^ ·K^−1^]	-
I	3.40	0.04	48.23	0.96	0.54	0.08	66.88	3.04
II	1.30	0.02	53.20	0.71	0.30	0.07	107.40	11.69
II + III	1.84	0.22	50.56	3.19	0.24	0.03	117.11	11.36
I + II + III	5.27	0.53	55.75	2.59	0.41	0.02	40.21	9.89

**Table 7 materials-15-05720-t007:** A summary of the parameters measured on the Alambeta device. Continuation.

Symbol of Knitted Fabric or Package	Thermal Resistance, r	Standard Deviation of Thermal Resistance σ_r_	Max and Stationary Heat Flow Quotient, I	Standard Deviation of Max and Stationary Heat Flow Quotient σ_I_	Stationary Heat Flow Density at Contact Point, s	Standard Deviation of Stationary Heat Flow Density at Contact Point σ_s_
	[10^−3^ W^−1^·K·m^2^]	-	-	-	[W·K·m^−2^]	-
I	72.71	0.04	5.72	0.39	0.51	0.04
II	28.50	0.02	1.80	0.14	0.40	0.04
II + III	36.86	0.22	1.92	0.29	0.39	0.10
I + II + III	92.45	0.53	5.45	0.66	0.35	0.05

## Data Availability

Data is contained within the article.

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
