# Peer review of "Design Methodology and Technology of Textile Footwear"

_materials, 2022, doi:10.3390/ma15165720_

Round 1

Reviewer 1 Report

The manuscript - materials-1812577 titled ‘’Design methodology and technology of textile footwear’’ has been reviewed. This paper describes a methodology for the design of sports and leisure textile footwear manufactured by innovative hybrid technology. The Air permeability and thermal insulation parameters were determined the different prepared materials.  The investigation is interesting but some improvements and corrections are required before the manuscript can be considered for publication in this journal. The following concerns are to be attended.

Recommendation:  Major revision

Comments:

1.      Page 12, allocate a number to the equation.

2.      I recommend the authors should provide ± standard deviation values for all data sets in the tables.

3.      Conclusions and future perspectives of the design methodology is high recommended to be outlined.

Reviewer 2 Report

Although, the manuscript is of interest to the readership of this journal, but authors to consider the following issues associated with the manuscript:

1. Method description should be isolated from Introduction section.

2. Machines and other related equipment should be described under Instrumentation section.

3. Section 3.1, Page 12, lines 280-281: Authors should state out the so 'calmed normal climatic condition' should be stated interns of a particular temperature and relative humidity.

4. Authors should number all Eq. in the manuscript including the one at line 283.

5. Figure 21: The suppose part of machine to be labeled 2 is missing, why?

6. Implications of the obtained results under mechanical properties of footwear upper should be discussed by the authors

7. English language should be enhanced throughout the manuscript.

Reviewer 3 Report

The manuscript is interesting.  However, some key issues must be properly addressed in details:   

1.       Introduction. Page 2, L46-50.  It is suggested to explain the backgrounds and objectives of this work in introduction section.  While material tests and experiments can be presented in later sections.  The current information and paragraphs should be restructured.  The current flow of information and the overall objectives are very confusing.

2.       Section 2 Materials and research methodology.  Pager 4, L114.  Why hybrid technology with embroidery is used, if the knitted fabric structure for footwear upper is superior and more durable than the traditional leather material (as stated in L67-68)?  What are the key variables being compared in this study? If knitted fabrics made of different structures are compared in this work?  Since the footwear making processes are highly complex, any chance to present the key processes in a flow diagram. 

3.       Section 3.  Testing textile materials used in sports footwear.  This section is irrelevant to the footwear making process in section 2.  What fabric samples are being tested?  Any reasons for the choice of yarns selected in this study?

4.       The tests should be repeated on leather upper for comparison purpose.

5.       Thermal insulation of materials should be conducted insider a chamber environment as the results could be easily affected by external environment and parameters.

6.        Limitations of this study?  Conclusion is missing.

Round 2

Reviewer 1 Report

Manuscript has been completely revised according to Reviewers' comments. My recommendation is accept for publication.

Author Response

The authors would like to thank the Reviewer 1.

Reviewer 2 Report

The quality of the manuscript has been enhanced.

Author Response

The authors would like to thank Reviewer 2 for his/her comments.